# Tailored Rehabilitation Program and Dynamic Ultrasonography After Surgical Repair of Bilateral Simultaneous Quadriceps Tendon Rupture in a Patient Affected by Gout: A Case Report

**DOI:** 10.3390/healthcare13151830

**Published:** 2025-07-26

**Authors:** Emanuela Elena Mihai, Matei Teodorescu, Sergiu Iordache, Catalin Cirstoiu, Mihai Berteanu

**Affiliations:** 1Department of Physical and Rehabilitation Medicine, Carol Davila University of Medicine and Pharmacy Bucharest, 050451 Bucharest, Romania; 2Department of Physical and Rehabilitation Medicine, Elias University Emergency Hospital, 011461 Bucharest, Romania; 3Department of Orthopedics and Traumatology, Carol Davila University of Medicine and Pharmacy Bucharest, 050451 Bucharest, Romania; 4Department of Orthopedics and Traumatology, Emergency University Hospital Bucharest, 050098 Bucharest, Romania

**Keywords:** bilateral quadriceps tendon rupture, simultaneous quadriceps tendon rupture, dynamic sonography, joint manipulation, home-based rehabilitation

## Abstract

Spontaneous quadriceps tendon rupture is a very rare occurrence, notably for bilateral simultaneous ruptures. Its occurrence is commonly linked to an underlying condition that may weaken the tendons leading to rupture. We report the case of a 68-year-old Caucasian male afflicted with long-term gout who presented a bilateral simultaneous quadriceps tendon rupture (BSQTR). We showcase the clinical presentation, the surgical intervention, rehabilitation program, dynamic sonographic monitoring, and home-based rehabilitation techniques of this injury, which aimed to improve activities of daily living (ADL) and quality of life (QoL). The patient was included in a 9-week post-surgical rehabilitation program and a home-based rehabilitation program with subsequent pain management and gait reacquisition. The outcome measures included right and left knee active range of motion (AROM), pain intensity measured on Visual Analogue Scale (VAS), functioning measured through ADL score, and gait assessment on Functional Ambulation Categories (FAC). All endpoints were measured at different time points, scoring significant improvement at discharge compared to baseline (e.g., AROM increased from 0 degrees to 95 degrees, while VAS decreased from 7 to 1, ADL score increased from 6 to 10, and FAC increased from 1 to 5). Moreover, some of these outcomes continued to improve after discharge, and the effects of home-based rehabilitation program and a single hip joint manipulation were assessed at 6-month follow-up. Musculoskeletal ultrasound findings showed mature tendon structure, consistent dynamic glide, and no scarring.

## 1. Introduction

Quadriceps tendon rupture is a rare injury, typically affecting men over the age of 50, with a male-to-female ratio of 8:1 [1]. Its incidence is estimated at 1.37 per 100,000 persons per year, reinforcing the rarity of the condition [1,2,3]. Additionally, bilateral tendon rupture is known to be even more uncommon and it comes with a greater disability in the affected individuals [2,3,4]. Most frequently, the lesion is linked to various systemic conditions, including diabetes mellitus, gout, chronic kidney disease, systemic lupus erythematosus, rheumatoid arthritis, connective tissue diseases, and hyperparathyroidism or risk factors such as obesity, quinolone, statins, or chronic steroid usage [2,5,6]. In some individuals, direct trauma can cause quadriceps tendon rupture [6]. Nevertheless, some cases have been reported in previously healthy individuals without any known risk factors [2,7].

Based on each individual’s susceptibility and biomechanical factors, in the event of a high suspicion of a simultaneous, bilateral quadriceps tendon rupture (BSQTR), prompt diagnosis, surgical treatment, and rehabilitation program are of paramount importance in order to achieve satisfying functional outcomes, by avoiding tendon retraction and quadriceps muscle atrophy that may occur with delayed repair [1,5,8]. The diagnosis is usually made through clinical assessment, revealing a palpable defect above the patella while the patient is unable to perform a straight leg raise, and magnetic resonance imaging (MRI) is the preferred choice as diagnostic tool [1,8,9]. However, it was reported that 67% of individuals were misdiagnosed at initial presentation, more frequently due to the rare occurrence of this type of injury [9]. In addition to MRI, dynamic musculoskeletal ultrasound plays a key role in both diagnosis and post-surgical monitoring, offering a non-invasive way to track tendon changes and muscle healing and guide individualized rehabilitation programs in real-time [10,11,12]. With high sensitivity (100% for quadriceps tendon, 87% for patellar tendon) and specificity, ultrasound is widely accessible in most clinical settings and complements clinical assessment effectively [10,11,12].

The authors present a case of bilateral, simultaneous rupture of the quadriceps tendon in a doctor afflicted with long-term gout, included in a post-surgery rehabilitation program with serial dynamic sonography, and home-based rehabilitation after discharge. Given the infrequent type of lesion, and its simultaneous, bilateral occurrence, we present the etiologic context, surgical protocol, and showcase the tailored rehabilitation program according to dynamic sonography to constantly assess in a non-invasive manner muscle architecture and tendons and adapt the rehabilitation program. This case is unique due to its bilateral nature and the use of dynamic sonography in rehabilitation. Given the scarcity of reports addressing bilateral quadriceps tendon ruptures—notably those combining serial dynamic ultrasonography with a personalized, staged rehabilitation protocol—this study fills an important gap in the literature. Although several case reports and series describe surgical techniques or basic rehabilitation strategies, few offer a comprehensive approach that incorporates imaging follow-up with functional adaptation and early return to independence. Therefore, the main goal of this case presentation is to provide guidance through the best surgical approach and suture type, rehabilitative strategies considering dynamic tracking of muscular and tendon changes, and ensure the best results by avoiding decondition after discharge through a tailored home-based rehabilitation program. Additionally, it also focuses on other interventions such as joint manipulation for better optimization of quadriceps muscle activity, notably post bilateral quadriceps tendon injury. This manuscript advances existing knowledge by demonstrating the utility of serial dynamic musculoskeletal ultrasound not only for diagnosis but also for monitoring healing progress and dynamically tailoring the rehabilitation process. Also, our approach involves outlining a structured rehabilitation protocol correlated with ultrasound findings, adaptable to the patient’s specific healing trajectory. Moreover, by illustrating a home-based rehabilitation strategy after discharge, we aimed to reduce healthcare burden while maintaining functional recovery. Another tailored strategy is using joint manipulation which helps highlight the role of this technique for muscle activation to ensure optimization of quadriceps muscle performance during recovery stage. Ultimately, this report contributes to improving clinical awareness, diagnostic accuracy, and functional outcomes in patients with rare, bilateral tendon injuries by promoting the integration of imaging and rehabilitation in a patient-specific, evidence-informed manner. The integration of combined rehabilitation protocols in everyday practice provides a comprehensive approach with the aim of achieving better recovery outcomes.

## 2. Materials and Methods

### 2.1. Case Presentation

The patient signed the informed consent form and agreed to take part in this case report, which was conducted in accordance with the Declaration of Helsinki and followed the CARE guidelines (for CAse REports). The IRB Committee approval was waived for this case report, as it is a single-case presentation. A 68-year-old Caucasian male was admitted to the Emergency Unit of “Elias” University Emergency Hospital, Bucharest, Romania, with functional impairment of both lower limbs along with bilateral gonalgia. The patient was employed as a doctor and denied previous knee pain or myalgias. He did not report any participation in regular contact sports and/or athletic activities.

The personal medical and surgical history analysis revealed the patient had type 2 diabetes treated with oral antidiabetic drugs, long-term gout, and hypertriglyceridemia. The patient’s body mass index (BMI) at admission was 26.23, indicating overweight status. The patient denied taking any corticosteroid medication and did not have any prior history of trauma. His general health was stable under standard treatment for the aforementioned conditions. While descending stairs, the patient misstepped and collided with another person, leading to a loss of balance. In an attempt to regain his stability, he rapidly loaded his flexed knees eccentrically—a mechanism that placed excessive tensile stress on the quadriceps tendons while they were contracting to decelerate the fall—ultimately resulting in bilateral tendon rupture. He reported hearing a “popping” sound, immediately followed by intense pain in the suprapatellar region bilaterally and with subsequent inability to stand without assistance. The clinical examination revealed swelling, tenderness, superiorly subluxated patella, suprapatellar depression, pain on the passive knee mobilization bilaterally, inability to perform any active straight leg raise and the extension lag sign positive. The deep tendon reflexes could not be assessed due to the painful palpation of both knees. Patient baseline status was stable; however, a high clinical suspicion of bilateral, simultaneous quadriceps tendon rupture was made during the clinical examination. Bilateral X-ray was performed the same day (Figure 1A,B). Due to limited availability of operating theatres, the patient was transferred to the University Emergency Hospital in Bucharest, Romania. Upon admission, MRI (Figure 1C,D) confirmed the clinical suspicion, revealing complete rupture of the quadriceps tendon at the level of the patellar insertion and quadriceps tendon retraction (Figure 1C,D).

### 2.2. Surgical Treatment

The patient was scheduled for surgery the 6th day after bilateral tendon rupture. He was positioned in the supine position and was under general anaesthesia. The anterior suprapatellar approach centred at the level of the quadriceps muscle with careful dissection of the anatomical planes, identification of the complete bilateral lesion of the quadriceps and evacuation of the remaining hematoma was used. The non-viable tissues were removed and the Krackow type of suture was applied at the level of the quadriceps and reinserted at the level of the upper pole of the patella using two anchors to fix the tendon to the bone (Arthrex SwiveLock) (Figure 2). The Krackow technique was selected based on its superior biomechanical strength and resistance to cyclic loading, as demonstrated in comparative suture studies. This method provides enhanced tendon-to-bone fixation and minimizes gap formation during early rehabilitation phases, supporting a safer and more effective recovery [1]. The patellar tracking was checked, the suture of the anatomical planes was carefully performed and a suction drain tube was introduced. Antibiotic prophylaxis with Cefuroxime was maintained for 48 h and both lower limbs were immobilized in a fixed orthosis, maintaining the full extension. The same surgical procedure was applied for left and right lower limbs.

Some challenges the surgical team had to consider were due to systemic conditions like diabetes and gout, as in the case of our patient; therefore, the possible degeneration of the tendons could have been an issue, making the tendon more friable and less able to hold sutures, but fortunately, the team successfully overcame these shortcomings. Additionally, given the bilateral involvement simultaneously, repairing both knees posed some ergonomic and logistical difficulties during the surgery, but the team managed to solve them effectively with a strict schedule and clear steps to be followed. Another challenge was achieving equal tension and symmetry in this bilateral repair to prevent postoperative gait imbalance or extensor lag. Also, the correct positioning of anchors as SwiveLock in the patella must be very precise to ensure strong fixation while avoiding patellar fracture.

### 2.3. Post-Surgical Rehabilitation Program

After the surgical intervention, on the 7th day, the patient was admitted to the Physical and Rehabilitation Medicine Department of Elias University Emergency Hospital, Bucharest, to start a post-operative rehabilitation program.

The clinical examination of the inferior limbs revealed post-surgical scars properly healed without signs of local inflammation, except for a slight increase in temperature (calor) of the right knee, which was autolimited. The atrophy of the quadriceps muscle as a result of disuse was also noted bilaterally. The post-surgery rehabilitation program consisted of three phases spread over a period of 9 weeks, during which the patient was closely monitored. An additional 4th phase started at discharge and consisted of a tailored home-based rehabilitation program. There are currently no universally accepted, specific rehabilitation guidelines dedicated solely to bilateral simultaneous quadriceps tendon rupture (BSQTR) due to its rarity. Most rehabilitation protocols are extrapolated from unilateral quadriceps tendon rupture rehabilitation protocols, clinical expertise, and case reports or case series. However, there are the key considerations and principles that guide rehabilitation for BSQTR, as we demonstrate in this work. The rehabilitation protocol we designed was also consistent with other case reports and evidence from the literature, and adapted in real-time according to the clinical assessment correlated with dynamic musculoskeletal sonography in the case of our patient [1,2]. Based on the particular findings of the case, along with existing literature, we guided the key phases, along with timing, goals, interventions and outcomes used to assess progression and the phased rehabilitation program, which is summarized in Table 1.

The outcome measures to guide progression included right and left knee active range of motion (ROM), pain intensity measured on the Visual Analogue Scale (VAS), activities of daily living (ADL), and gait assessment on Functional Ambulation Categories (FACs). All endpoints were measured at different time points (T0, T1, T2, T3, T4—Table 2). The ADL score is a standardized tool used to assess an individual’s ability to perform basic self-care tasks necessary for independent living as it mainly evaluates physical functioning. The score includes core daily functions such as bathing, dressing, using the toilet, transferring, continence and feeding, and each activity is rated based on the degree of independence, with scores indicating either level of independence, need for assistance, or dependence. It is a valuable tool to help track recovery or decline in function and it is assessed either through observation, self-report, or caregiver interviews.

The first phase (weeks 1–4) focused on pain therapy and management, wound healing, the protection of the surgical repair (both knees hold with braces locked in full extension), maintenance of overall joint mobility, the preservation of overall muscle strength, and support of the cardiovascular system and function. Additionally, low-level laser therapy (LLLT) was applied to optimize and complement rehabilitation outcomes. In the second phase (weeks 5–7), we continued protecting the surgical repair (both knees with braces locked in full extension (up to week 6)) (Figure 3A), focused on the regain of the knee motion and amplitude, began muscle strengthening exercises, and focused on the cardiovascular function with gait adaptation and normalization. Passive mobilization was initiated with the aim of increasing the flexion angle within the painful threshold. The third phase (weeks 8–9) (Figure 3B) focused on regaining the knee range of motion (ROM), quadriceps muscle strengthening, gait normalization both on even and uneven surfaces, regaining independence in basic ADLs and preparing the return to home and work. The last phase consisted of a tailored home-based rehabilitation program with a list of exercises explained and a printed handout. The focus was on social reinsertion, return to work, mild sport activities and no contact sports.

### 2.4. Dynamic Sonography for Optimizing Post-Rehabilitation Program

Serial dynamic musculoskeletal ultrasound assessments were performed at admission and at 6 and 9 weeks postoperatively by a single experienced physiatrist to ensure consistency and eliminate inter-rater variability. Standard musculoskeletal ultrasound settings were applied, with the gain adjusted between 40 and 60 dB to optimize tendon and muscle visualization in both longitudinal and transverse views during dynamic assessment of the quadriceps tendons. Moreover, along with other physiatrists who also were part of the rehabilitation team of our patient, based on clinical and ultrasound findings, we made real-time decisions for the rehabilitation program. The reasoning behind dynamic musculoskeletal ultrasound is that this valuable tool is non-invasive, repeatable, and enables real-time feedback to adjust rehabilitation based on objective findings such as tendon continuity and healing status, structural integrity (e.g., fibrillar pattern, scarring), muscle activation during motion, tendon glide and contractility, effusion or post-surgical complications (e.g., hematoma, adhesions) [10,11,12]. All these features were taken into consideration for every step of the rehabilitation program and helped adapt the program in real-time, allowing a tailored intervention according to the patient’s post-surgical and rehabilitation stage. Literature findings highlight novel ultrasound-guided rehabilitation strategies and demonstrate how imaging can be used dynamically to guide treatment and decision-making, reinforcing the role of ultrasonography in musculoskeletal rehabilitation pathways [13]. A literature review contextualizes ultrasound findings in chronic tendinopathies and supports the utility of imaging follow-up and real-time adjustments in accordance with tendon and muscle architecture changes, while reinforcing our approach towards dynamic sonographic monitoring in the case of our patient, notably in chronic conditions such as long-term gout and diabetes, which frequently impact tendon structure [14]. The rehabilitation program was tailored according to muscle and tendon appearance, inflammation and recovery, aiming to provide a real-time strategy adapted to the musculoskeletal needs of the patient for each rehabilitation phase. Figure 4 highlights different bilateral sonographic assessments of quadriceps tendon at different time points (T0, T1, T2) presenting architectural and inflammatory changes. At baseline assessment (T0), the quadriceps tendon had lost the fibrillar aspect and demonstrated a thickened pattern as well as the presence of edema. According to the sonographic findings, the rehabilitation program focused on pain management, wound healing, surgical protection, the maintenance of ROM, muscle strength, and overall function. LLLT was also applied to decrease inflammation and assist healing. The musculoskeletal ultrasound at 6 weeks post-surgery revealed a relatively homogeneous, echogenic left and right quadriceps tendon with fibrillar pattern, several peritendinous calcifications and reduced subcutaneous tissue edema compared to the baseline evaluation. Correlating these findings with the clinical assessment, we started the second rehabilitation phase (weeks 5–7), and the braces were removed at week 6 post-surgery. We continued the passive mobilization and also began active muscle strengthening. By the end of the post-surgical rehabilitation program (week 9), the quadriceps tendon had regained homogeneity and its fibrillar aspect and the subcutaneous tissue edema was resorbed. Therefore, progressive strengthening, walking on even and uneven surfaces, and stair training were initiated for our patient. We aimed to use serial ultrasound to better adapt the post-surgical rehabilitation plan to the patient’s needs while considering the tendon appearance and edema level. Considering the ultrasonographic changes of tendons and muscles and adapting the rehabilitation program accordingly led to setting active goals in rehabilitation and facilitate recovery through increased speed rate and better outcomes at discharge and follow-up.

Knee injuries, affecting both young individuals and adults, are a prevalent issue in musculoskeletal health [1,2]. For example, reduced quadriceps strength was associated with an increased risk of further cartilage deterioration in the lateral patellofemoral joint in women. This implies that improving quadriceps strength could potentially help prevent the progression of structural damage in the patellofemoral joint in women [4]. One of the key factors driving this degeneration is the weakening of the quadriceps femoris (QF) muscle, a critical muscle group that plays a central role in maintaining knee stability and proper biomechanics [5].

### 2.5. Single Lateral Hip Joint Manipulation at Six Months Considering Better Biomechanical Support and Neuromuscular Aspects

After a tailored rehabilitation program during hospital stay and after discharge, for our patient some persistent biomechanical deficits were obvious—including poor single-leg stance control, delayed quadriceps recruitment, and compensatory trunk sway—that warranted a proximal joint intervention to re-establish more optimal joint kinematics and neuromotor timing. By restoring hip mobility and reducing soft tissue restrictions, the manipulation likely enhanced the patient’s capacity for safe and symmetrical lower limb loading during gait and therapeutic tasks. Therefore, we proposed physical therapy management using a single hip manipulation and therapeutic exercises six months after quadriceps tendon rupture surgery and rehabilitation program. This intervention was based on the concept of regional interdependence, whereby optimizing hip joint mobility and lumbopelvic neuromotor control can indirectly enhance knee function and lower limb biomechanics. Hip manipulation has been shown to temporarily increase motor excitability and activation of muscles along the kinetic chain, particularly the vastus medialis and surrounding stabilizers of the knee [15]. In support of this, a recent systematic review and meta-analysis by Runge et al. found that adding manual therapy to exercise therapy significantly improves pain and functional outcomes in patients with knee and hip osteoarthritis—underscoring the neuromuscular and biomechanical benefits of these techniques beyond passive symptom relief [16]. The patient was informed about the intervention and provided with existing evidence. Before the procedure, the patient had a previous assessment with the therapist and scheduled a new appointment for hip manipulation. An experienced therapist performed the procedure with no difficulty and no pain for the participant. The supine position with a 30° flexion at the hip joint and upper limbs resting alongside the trunk was used. The therapist performed a lateral thigh movement, parallel to the ground, until the maximum felt resistance of the tissue tension in the hip joint was reached. After making sure that the patient did not feel any discomfort or pain, a 1 s lateral manipulation was performed. One week after the intervention, the patient reported better knee joint stability and a better level of comfort in his activities. To objectify this, pre- and post-manipulation, the following tests were carried out to assess patient’s strength and stability: Star Excursion Balance Test (SEBT) to evaluate dynamic postural control and proprioceptive stability given its usefulness in detecting unilateral or bilateral biomechanical deficits; Single-Leg Stance Time to assesses postural control, gluteal and quadriceps stability as improvement may correlate with perceived stability and neuromuscular reactivation; Timed Up and Go (TUG) to measure lower limb function and balance. Improvements after manipulation may reflect better neuromuscular efficiency.

However, clinicians must consider contraindications before applying joint manipulations. These include hip joint instability, osteoporosis, coagulopathies, previous hip arthroplasty, or any signs of metastatic involvement. In this case, thorough screening excluded these risks, and the manipulation was well tolerated without adverse effects.

## 3. Discussion

To the best of our knowledge, our case study is the first to describe the full steps taken from high suspicion to the diagnosis of bilateral, simultaneous quadriceps tendon rupture, to post-surgery tailored rehabilitation correlated with dynamic sonography of the muscles and tendons, home-based rehabilitation program after discharge, and joint manipulation. The timely multidisciplinary, tailored approach aimed to provide a comprehensive strategy in order to ensure and facilitate the return to ADLs, work, social activities and sports, attaining the best functional outcomes and aiming for a satisfying quality of life. Moreover, the patient was able to return to activities of daily living and work upon 12 weeks after surgery and 2 weeks after discharge from the post-operative rehabilitation program. Some case reports showed both similar or slightly different results on ADLs [1,5,17].

In the literature, slightly more than 100 cases of bilateral quadriceps tendon ruptures have been reported [2]. The effect of a complete tear of the quadriceps tendon ends in loss of muscle function and inability to maintain an upright position or walk [8]. More frequently, this is a simultaneous occurrence after a fall or traumatic injury and it is thought that the forces needed to lead to quadriceps tendon rupture must be very vigorous, given the strength of the structures and their resistance to heavy loads [2,4]. While bilateral quadriceps tendon rupture is rare, clinicians should remain vigilant in patients presenting with sudden loss of knee extension following minor trauma, particularly in those with metabolic comorbidities [1,9]. Additionally, males are also more prone to experiencing a complete tear [2,18]. Gout, along with diabetes or other conditions, as presented in the literature and seen in this case, can predispose tendons to rupture due to chronic crystal deposition, local inflammation, and associated degenerative changes that weaken tendon integrity over time, even in those individuals who do not engage in contact sports or strained physical activity [2,5,6]. Warning signs include persistent anterior knee pain, reduced extensor strength, and a palpable suprapatellar gap. Early imaging—especially ultrasound or MRI—should be pursued in high-risk patients to confirm diagnosis [1,8,9,10,11,12]. This case underscores the importance of recognizing subtle tendon pathology in at-risk individuals and tailoring multidisciplinary care accordingly.

In the case of our patient, the anamnesis and clinical examination raised high suspicion for simultaneous, bilateral quadriceps tendon rupture and the diagnosis of tendon rupture is mainly based on clinical findings. However, the gold standard for the diagnosis of tendon ruptures is magnetic resonance imaging (MRI) [9,12,19]. MRI offers a thorough anatomic view and great soft tissue contrast, being of significant help for pre-operative reconstructive planning [1,20]. To complement the clinical findings and the assessment of the muscles and tendons in real time, ultrasound is a great non-invasive and economical method [11]. The main goal should be early diagnosis and surgical repair, leading to results comparable to unilateral tendon ruptures [19]. In the case of partial ruptures, management can remain non-operatively, while surgical repair is usually applied for complete ruptures of quadriceps tendons [6]. There are several surgical reparatory techniques that have been described as the treatments of choice: direct repair using Bunnel or Krackow sutures and suture anchors in acute cases or repair using lengthening techniques by applying grafts in chronic cases [1,13]. Reconstructive surgery performed in the acute state aims to ensure full regain of muscle and tendon strength and function [21]. Even though there are various surgical techniques and sutures that can be used for the treatment of a tendon rupture, the surgical team considered the biomechanical properties of the tendons and opted for the Krackow suture, whereas some studies have shown that the Krackow and the Bunnel type of sutures are superior to the Kessler suture when it comes to factors such as strength, endurance, and also durability [22,23]. Opting for a Krackow suture can be more challenging due to the technique, but this type of suture presented a more prominent axial resistance than the other types of sutures [24].

Various studies have shown that the earlier the surgical repair and physical therapy are performed, the better the functional outcomes compared to delayed repair of the quadriceps muscle [25,26]. Additionally, a delay in diagnosis and reparatory surgery may frequently lead to more surgical interventions and is linked to poorer post-surgical functional outcomes, more persistent knee extensor weakness, decreased range of motion and increased risk for complications [19,20,21,27]. In the case of our patient, the surgery was conducted in the first 6 days after the lesion, after considering all the risk factors and comorbidities, complications, and outcomes.

A post-surgery rehabilitation program is highly required to optimize and ensure a satisfying functional outcome and quick reinsertion into activities of daily living (ADL) [1]. In the post-operative rehabilitation program, we opted for dynamic sonography for our patient aiming to provide real-time, tailored rehabilitation techniques adapted to the needs of our patient, which represents a new approach regarding the management of quadriceps tendon rupture [13,14]. Additionally, the medical management focused on the treatment of gout and adaptation of diabetic oral medication. Subsequently, the optimization of nutrient intake and avoidance of foods that may increase uric acid and urea were attentively applied. An early post-surgical rehabilitation program is of paramount importance in attaining quick recovery and satisfying functional outcomes [1,28]. In the case report we present, the patient started a well-adapted and structured rehabilitation program 5 days per week, twice daily, during a period of 9 weeks, as it seemed that the number of physical therapy sessions was directly related to an individual’s improvement of functional outcomes. Evidence showed that individuals who attained excellent functional outcomes attended more therapy sessions compared to individuals who attained good, fair, or/and pair functional outcomes [29]. Additionally, a full return to a previous competitive level is shown to be related to an all-encompassing, goal-oriented rehabilitative program, where patients had access to daily physical therapy sessions and activities [1,28]. In the case of our patient, we adapted the rehabilitation program in real time according not only to the dynamic sonography, but also based on overall performance to physical therapy sessions and adhesion to medical and rehabilitation recommendations. Furthermore, low-level laser therapy (LLLT) was applied aiming to enhance the effects of physical therapy sessions, help the body to regenerate more quickly and attain a good level of relaxation, notably when the patient reported higher pain scores that interfered with the physical therapy sessions [30,31]. Although there is still the need for larger patient groups, some studies have shown promising results regarding the usage of LLLT on tendons and evidence indicated that LLLT has an important effect on tendon repair [30,31]. LLLT activates cytochrome C oxidase and carries out the photon absorption process, is involved in all three phases of tendon repair, and is responsible for improving the recovery of the tendons [30]. LLLT promotes tendon healing, decreases pain intensity, improves the flexibility of soft tissue, augments joint mobility, and therefore leads to higher patient compliance to the rehabilitation program [31].

Given the course of healing after surgery and patient adherence to the rehabilitation program during the hospital stay and after discharge, after careful evaluation and discussion with the patient, we also took into consideration the application of joint manipulation. Despite good results in terms of ROM and pain intensity, the patient still felt knee instability, which could be explained by long muscle disuse and deconditioning. A growing body of evidence has shown that hip joint manipulation has a positive impact in knee injuries on biomechanical and neuromuscular levels, and given our patient’s history, secondary prevention became crucial. Data have shown that training of the quadriceps can significantly reduce knee pain, and many rehabilitation programs have started to increasingly focus on strengthening the quadriceps muscle to restore knee function [32,33]. The quadriceps muscle plays a crucial role in stabilizing the knee joint during movement; therefore, any strength deficit or poor neuromuscular control can significantly impact knee biomechanics, leading to instability [33,34,35]. Neuromuscular impairments, including inhibition or abnormal activation of the quadriceps muscle, can lead to symptoms that alter the normal function of the joint, increasing the risk of additional injury [33,36]. Adapted neuromuscular control is crucial for achieving active joint stability, ensuring safe and efficient movements in activities of daily living, as well as during training [33]. Recent data have highlighted the potential of manual therapy to enhance neuromuscular function [33,37]. Joint manipulation, usually characterized by thrusts with high velocity and low amplitude, is mostly used for joint pain or restricted range of motion [33,38]. The effects are present beyond the biomechanical changes, acting also at the neurophysiological and supraspinal level, therefore, impacting muscle function and neuromuscular control [33,38]. A randomized controlled trial (RCT) demonstrated the efficacy of high-velocity and low-amplitude hip manipulation in enhancing muscle activation, notably for vastus medialis and vastus lateralis muscles, showing promising results by adding this type of intervention into rehabilitation programs to help improve outcomes for individuals recovering from knee injuries or even for those interested in sports [33]. It is important for practitioners to consider the roles of these muscles in stabilizing the knee as well as the extension when tailoring rehabilitation programs. Evidence suggests that high-velocity and low-amplitude hip manipulation can have an effect on muscle modulation and quadriceps activation, leading to optimization of the neuromuscular function with beneficial effects on knee injuries, pain, and stability [33]. One week after one session of hip manipulation, the patient self-reported better stability of the knee joint, as well as more ease in carrying out daily life activities and during work tasks.

The decision to maintain the knee joint immobilized in full extension for 4 to 6 weeks after reparatory surgical treatment was based on evidence that showed that this amount of time is the best approach to protect tendon repair and ensure complete healing [1,21]. In our case report, both of the patient’s knees were immobilized in a functional brace for 6 weeks in full extension with a subsequent slow degree increase in flexion up to week 7, when the knee brace was finally removed. At discharge, the patient was able to ambulate on even and uneven terrain safely, without crutches or braces. Different from other case reports, we also opted for a tailored home-based rehabilitation program, which consisted of muscle-strengthening exercises, walking sessions and focused on dynamic balance. Additionally, considering biomechanical and neuromuscular aspects, a single hip joint manipulation was also performed for our patient. Attention to diet and medical conditions was also a very important part, considering that gout is linked to quadriceps tendon rupture along with other conditions and risk factors. Therefore, secondary prevention plays a crucial role, notably for our patient.

The long-term rehabilitation goals were centered on restoring lower limb strength, functional independence, gait stability, and reintegration into daily and occupational activities. These objectives were pursued through a structured inpatient and home-based rehabilitation program, closely guided by serial dynamic musculoskeletal ultrasound assessments to tailor interventions in real-time. Given the patient’s residual deficits, particularly persistent quadriceps weakness and altered lower limb biomechanics, additional therapeutic strategies were explored. Six months post-surgery, a single high-velocity, low-amplitude (HVLA) hip manipulation was incorporated into the treatment plan to address compensatory pelvic alignment issues and facilitate neuromuscular activation of the quadriceps. Such targeted interventions are consistent with a global, individualized rehabilitation approach aimed at optimizing recovery, addressing biomechanical impairments, and promoting long-term stability and function.

In the case of our patient, dynamic musculoskeletal ultrasound was performed exclusively by a skilled physiatrist who also managed the patient’s care, ensuring consistency and continuity in assessment. Real-time ultrasound imaging allowed for objective monitoring of tendon healing, including continuity, fibrillar structure, muscle activation, tendon glide, and detection of complications such as hematomas or adhesions. These ultrasound findings, combined with clinical evaluation and informed ongoing rehabilitation decisions, enabled the interdisciplinary team to adapt the therapeutic plan in real-time. This individualized, responsive approach enhanced the precision and safety of each rehabilitation phase, optimizing patient recovery. Our approach towards dynamic musculoskeletal ultrasound is also consistent with literature findings. A narrative review highlights novel ultrasound-guided rehabilitation strategies, demonstrating how imaging can be used in a dynamic manner to guide treatment and reinforcing the growing role of ultrasonography within musculoskeletal rehabilitation programs [13]. By contextualizing ultrasound findings in chronic tendinopathies, it further supports the utility of imaging follow-up, complementing the usage of dynamic ultrasonography for personalized, real-time rehabilitation adjustments as in the case of our patient, helping to set active goals [14].

This case report has several strengths and limitations as well. One strong feature is comprehensive and individualized care: the case illustrates a well-structured, phase-based rehabilitation program integrating clinical, imaging, and functional assessments, including dynamic ultrasound monitoring, home-based rehabilitation, and joint manipulation. Additionally, the use of serial musculoskeletal ultrasound enabled real-time tracking of tendon healing, allowing tailored rehabilitation interventions based on healing progress. The integration of home-based rehabilitation ensured continuity of care and patient empowerment after discharge, therefore promoting functional independence and social reintegration. Another strength is the innovative inclusion of hip manipulation, as the case report contributes to emerging evidence suggesting that manual therapy may enhance neuromuscular activation, especially in the context of chronic muscle weakness. Limitations are related to the single-case design, which limits generalizability and outcomes may not apply to broader patient populations or different healthcare systems and medical contexts. Additionally, the follow-up only extended to six months after surgery, but long-term tendon integrity, return to sports, work, and quality of life may still suffer changes after this period of time. Also, other alternative treatments and interventions that might have been suitable as well (e.g., continuous passive range of motion, neuromuscular reeducation devices) were not explored for comparison.

### Take-Home Messages

Prompt referral and MRI confirmation can reduce misdiagnosis and improve surgical timing and outcomes;Timely surgical intervention is essential to prevent tendon retraction and irreversible muscle atrophy;A dynamic and individualized rehabilitation program guided by serial dynamic musculoskeletal ultrasound and thorough clinical evaluation ensures that therapy adapts to the patient’s healing trajectory; continuous monitoring supports tissue healing, muscle reactivation, and safe progression in mobility, strength, and function;Home-based and long-term recovery strategies extend functional gains beyond inpatient care, promoting independence and reducing healthcare burden;The role of manual therapy in functional reintegration after a single high-velocity, low-amplitude hip manipulation, strategically applied at 6 months post-surgery, may facilitate neuromuscular activation and improve biomechanical alignment; when used selectively, manual therapy can enhance proprioception and support return to ADLs;Active goal-setting and multidisciplinary involvement: engaging the patient in active goal-setting fosters motivation and promotes recovery; close collaboration between surgical and physical and rehabilitation medicine teams enables comprehensive care that addresses strength, function, and stability.

## 4. Conclusions

This case highlights the importance of timely surgical intervention combined with a structured, multidisciplinary rehabilitation strategy for optimal recovery following bilateral quadriceps tendon rupture. The integration of dynamic musculoskeletal ultrasound throughout rehabilitation enabled individualized, real-time adjustments that supported tendon healing and functional progress. A well-supervised home-based program helped sustain these gains after discharge, ensuring a safe and smooth reintegration into social and professional life. Notably, the addition of a targeted single hip manipulation contributed to enhanced quadriceps activation and neuromechanical stability, addressing persistent biomechanical deficits that may not fully resolve despite surgical repair and rehabilitation programs. This tailored and comprehensive approach underscores the value of combining surgical, rehabilitative, and biomechanical strategies to optimize outcomes. The comprehensive management plan presented here offers a useful model that can be applied to similarly complex musculoskeletal injuries. Broader adoption of such individualized protocols may enhance long-term function, restore independence, and improve quality of life in active patients recovering from major tendon injuries.

## Figures and Tables

**Figure 1 healthcare-13-01830-f001:**
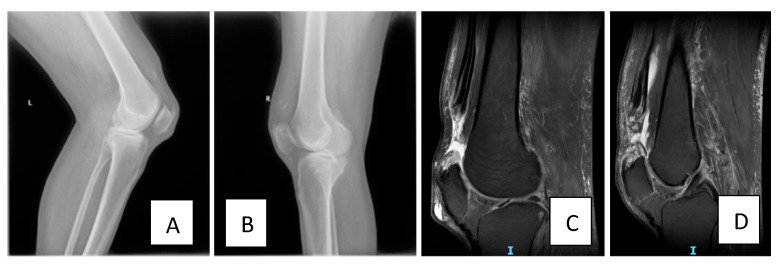
(**A**) Left knee lateral view on X-ray (preoperative, performed immediately after the incident) showing the low position of the patella. (**B**) Right knee lateral view on X-ray showing the low position of the patella (preoperative, performed immediately after the incident). (**C**) Sagittal T2 left knee MRI image (preoperative, performed 1 day after the incident) showing complete rupture of the quadriceps tendon at the level of the patellar insertion with 40 mm retraction. (**D**) Sagittal T2 right knee MRI image (preoperative, performed 1 day after the incident) showing complete rupture of the quadriceps tendon at the level of the patellar insertion with 24 mm retraction cranially.

**Figure 2 healthcare-13-01830-f002:**
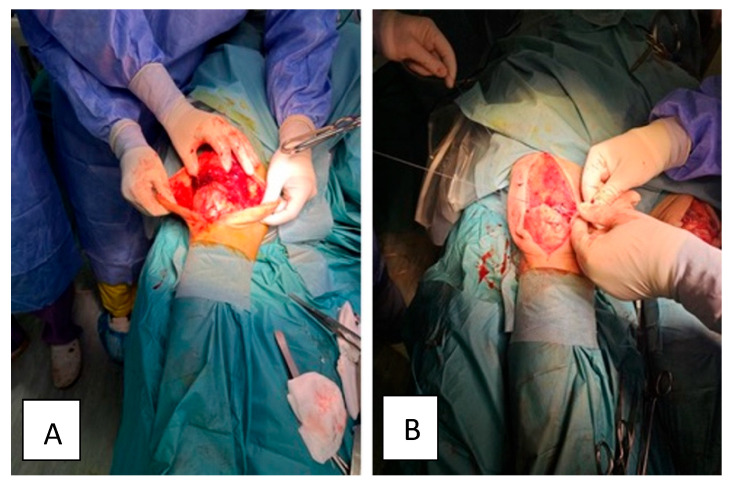
(**A**) Intraoperative image showing complete rupture of the quadriceps tendon. (**B**) Intraoperative image showing the reinsertion of the quadriceps at the level of the upper pole of the patella using fixation anchors.

**Figure 3 healthcare-13-01830-f003:**
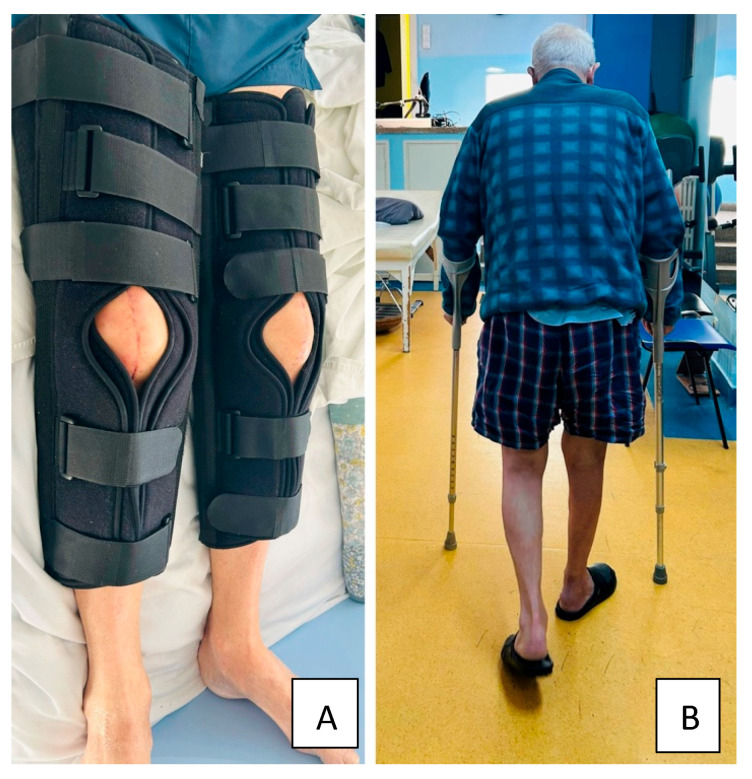
(**A**) Patient wearing a bilateral functional brace (unhinged ROM knee brace adjustable with extended support) for 6 weeks to keep the lower limbs in full extension (the picture was taken at 5 weeks post-surgery). (**B**) Patient walking with the aid of crutches at 8 weeks post-surgery on even surface and without crutches at discharge (9 weeks).

**Figure 4 healthcare-13-01830-f004:**
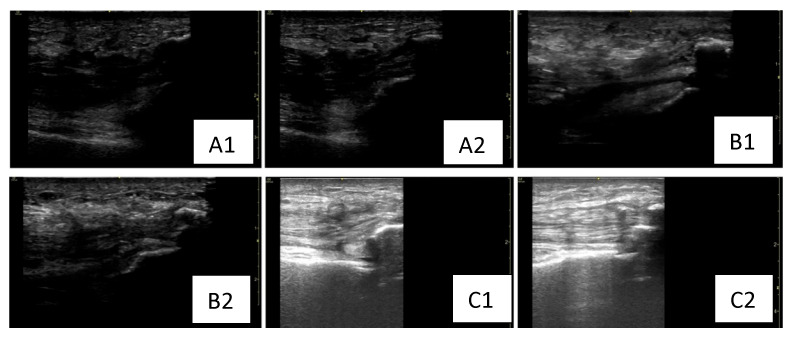
Serial dynamic sonography for left and right lower limbs assessed at different time points (T0, T1, T2) post-surgery. (**A1**,**A2**) Relatively non-homogeneous, hypoechoic left and right quadriceps tendon with loss of fibrillar pattern and thickened appearance, associated with the presence of peritendinous calcifications and subcutaneous tissue edema (baseline evaluation). (**B1**,**B2**) Relatively homogeneous, echogenic left and right quadriceps tendon with fibrillar pattern, several peritendinous calcifications and reduced subcutaneous tissue edema (sonographic evaluation at 6 weeks post-surgery). (**C1**,**C2**) Homogeneous, hyperechoic left and right quadriceps tendon with fibrillar pattern, presenting several peritendinous calcifications and resorption of the subcutaneous tissue edema (sonographic evaluation at 6 weeks post-surgery). T0: baseline sonographic assessment at admission; T1: sonographic assessment at 6 weeks post-surgery.; T2: sonographic assessment at 9 weeks post-surgery (at discharge).

**Table 1 healthcare-13-01830-t001:** Phased rehabilitation program.

Phase	Timeline	Key Goals	Interventions	Measurements
Phase 1	Weeks 1–4	Pain management; wound healing and surgical protection; maintain ROM, muscle strength, and overall function	Knee braces locked in full extension; LLLT therapy; Cardiovascular support; Passive ROM; Muscle activation.	VAS, ROM, ADL, FAC
Phase 2	Weeks 5–7	Regain motion; initiate muscle strengthening exercises; cardiovascular rehabilitation	Braces (until week 6); Passive mobilization; Begin active strengthening;	VAS, ROM, ADL, FAC
Phase 3	Weeks 8–9	Normalize gait; increase strength and ROM; regain ADL independence	Progressive strengthening; Walking on even and uneven surfaces; Stair training.	VAS, ROM, ADL, FAC
Phase 4	Week 10 onward	Social and work reintegration; sport resumption	Tailored home-rehabilitation program; Printed handout; Mild aerobic exercises.	Follow-up ROM; ADL; return-to-work and non-contact sport activities

Abbreviations: ADL: activities of daily living; FACs: Functional Ambulation Categories; LLLT: low-level laser therapy; ROM: range of motion; VAS: Visual Analogue Scale. The structured rehabilitation program’s sessions were held 5 days per week, twice daily, during a period of 9 weeks. LLLT was applied once daily after surgical scars healing, for 6 weeks. The parameters used for LLLT were the following: wavelength: 808 nm; power: 200 mW; spot size: 0.04 cm^2^; energy per point: 6 J.

**Table 2 healthcare-13-01830-t002:** Outcome measures evaluated at different time points: T0, T1, T2, T3 and T4.

Outcome Measures	T0	T1	T2	T3	T4
Right Knee AROM (degrees)	0	25	55	95	Full
Left Knee AROM (degrees)	0	20	60	90	Full
VAS	7	4	3	1	0
FAC	0	0	1	5	5
ADL	6	6	7	10	10

Abbreviations: AROM: active range of motion; FACs: Functional Ambulation Categories; VAS: Visual Analogue Scale. T0: baseline assessment at admission; T1: assessment at 4 weeks; T2: assessment at 6 weeks; T3: assessment at 9 weeks (discharge); T4: follow-up assessment 6 months after discharge, home-based rehabilitation and hip joint manipulation.

## Data Availability

Data are contained within the article.

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
