# Peer review of "Tailored Rehabilitation Program and Dynamic Ultrasonography After Surgical Repair of Bilateral Simultaneous Quadriceps Tendon Rupture in a Patient Affected by Gout: A Case Report"

_healthcare, 2025, doi:10.3390/healthcare13151830_

Round 1
Reviewer 1 Report
Comments and Suggestions for Authors
- In Table 1, please provide additional details regarding the ADL score—specifically, what it measures and how it is assessed.
- Please expand the discussion on the significance of using ultrasonography. While the results indicate findings such as calcification and edema, it is unclear how these findings influenced the rehabilitation protocols.
Author Response
Dear Editor and Reviewers,
On behalf of our entire research team, we extend our gratitude for your invaluable assistance and guidance in refining our manuscript. Your insightful suggestions and expert feedback have been extremely helpful in enhancing the quality and clarity of our work. We have thoroughly addressed every query, incorporating substantial revisions that we believe significantly strengthen our paper. Once again, we express our appreciation for your dedicated review and time.
As a testament to our commitment to addressing your feedback thoroughly, we have enclosed two documents: the comprehensively revised manuscript, with modifications highlighted for easiness to find, and a comprehensive response document detailing our point-by-point responses to your insightful comments and recommendations.
Should there be any further points requiring our attention, we assure you that we remain fully committed to resolving any remaining issues to the best of our abilities. We are highly appreciative and grateful for the time and effort you have dedicated to our work.
Below you can find our point-by-point responses.
Thank you.
Kindest regards,
Emanuela Elena Mihai, M.D., Ph.D.
Reviewer 1 :
- In Table 1, please provide additional details regarding the ADL score—specifically, what it measures and how it is assessed.
Author`s reply: Thank you for your comment and for pointing this out. We have provided additional details in the manuscript highlighted in red regarding the ADL score- specifically about the measurements and the way it was assessed. The following paragraph was added in the manuscript according to your recommendation:
“The ADL score is a standardized tool used to assess an individual's ability to perform basic self-care tasks necessary for independent living as it mainly evaluates physical functioning. The score includes core daily functions such as: bathing, dressing, toileting, transferring, continence, feeding, and each activity is rated based on the degree of independence, with scores indicating either level of independence, need for assistance, or dependence. It is a valuable tool to help track recovery or decline in function and it is assessed either through observation, self-report, or caregiver interviews.”
- Please expand the discussion on the significance of using ultrasonography. While the results indicate findings such as calcification and edema, it is unclear how these findings influenced the rehabilitation protocols.
Author`s reply: Thank you for your input. We have expanded the discussion section accordingly on the significance of using ultrasonography and how these findings influenced the rehabilitation protocol. The following paragraph was added in the manuscript according to your recommendation:
“Serial dynamic musculoskeletal ultrasound assessments were performed at admis-sion, and at 6 and 9 weeks postoperatively, by a single experienced physiatrist to ensure consistency and eliminate inter-rater variability. Standard musculoskeletal ultrasound settings were applied, with the gain adjusted between 40–60 dB to optimize tendon and muscle visualization in both longitudinal and transverse views during dynamic as-sessment of the quadriceps tendons. Moreover, along with other physiatrists who also were part of the rehabilitation team of our patient, based on clinical and ultrasound findings, we made real-time decisions for the rehabilitation program. The reasoning behind dynamic musculoskeletal ultrasound is that this valuable tool is non-invasive, repeatable, and enables real-time feedback to adjust rehabilitation based on objective findings such as tendon continuity and healing status, structural integrity (e.g., fibrillar pattern, scarring), muscle activation during motion, tendon glide and contractility, effusion or post-surgical complications (e.g., hematoma, adhesions). All these features were taken into consideration for every step of the rehabilitation program and helped adapt the program in real-time, allowing a tailored intervention according to patient post-surgical and rehabilitation stage.”
Reviewer 2 Report
Comments and Suggestions for Authors
Comments on the manuscript “Tailored Rehabilitation Program and Dynamic Ultrasonography after Surgical Repair of Bilateral Simultaneous Quadriceps Tendon Rupture in a Patient Affected by Gout: A Case Report” with manuscript ID: healthcare-3728528
The manuscript presents a well-documented case of bilateral simultaneous quadriceps tendon rupture in a patient with gout, detailing the surgical repair, rehabilitation program, and dynamic sonography monitoring. The study is clinically relevant, especially given the rarity of such cases, and provides valuable insights into tailored rehabilitation approaches. However, some sections could benefit from clearer organization, more concise language, and additional details to enhance readability and scientific accuracy.
Abstract
- It would be helpful to include specific quantitative outcomes (e.g., ROM, VAS scores) to highlight the effectiveness of the interventions.
- Consider rephrasing "aiming to improve activities of daily living (ADL) and quality of life (QoL)" to "which aimed to improve ADL and QoL" for smoother flow.
- Spell out abbreviations on first use (e.g., ADL, QoL).
- Add one sentence on objective ultrasound findings at follow-up to balance the functional outcomes already described.
Introduction
- The reason for presenting this case is well established. However, the introduction could be clearer in stating the manuscript’s objectives and explaining how it advances existing knowledge.
- The epidemiology and rationale are well explained, but the opening paragraph contains many statistics in a single sentence, which makes it harder to read. Splitting it up and adding one or two recent references (post-2022) would improve the context.
- The introduction offers a solid background on quadriceps tendon ruptures and their links to systemic conditions. However, the transition to the case report could flow more smoothly. Consider adding a sentence that explicitly highlights the novelty of this case, such as "This case is unique due to its bilateral nature and the use of dynamic sonography in rehabilitation."
- The paragraph on ultrasonography is informative but could be shortened to reduce redundancy with later sections.
Case Presentation
- Consider providing more information on the patient’s baseline functional status and comorbidities, as this would help contextualize the rehabilitation outcomes.
- The clinical narrative is detailed and flows logically. You might shorten the transfer-of-care details to maintain momentum and include the patient’s body mass index, which is often important in tendon issues.
- The injury mechanism could be explained in more detail. For example, how did the misstep cause maximum knee flexion? A short biomechanical explanation would improve this section.
- The MRI findings are clear, but the figure labels (A, B, C, D) are not referenced in the text. Refer to the figure when describing the images.
Surgical Treatment
- The surgical procedure is described in detail, but the rationale for choosing the Krackow suture over other techniques could be expanded. Referring to biomechanical studies (e.g., suture strength comparisons) would add depth to the discussion.
- Mention any intraoperative challenges or unexpected findings, if applicable, to provide a fuller picture of the surgical experience.
- The phased rehabilitation program is comprehensive, but the timeline could be visualized in a table or flowchart for clarity.
Post-Surgical Rehabilitation Program
- The phased program is a strength of the report. To enhance reproducibility, consider adding details on frequency and intensity for key exercises and clarifying the dosage of low-level laser therapy. A concise table with phase goals, precautions, and progression criteria would assist clinicians in replicating the regimen.
Dynamic Sonography
- Including representative ultrasound images (if permitted) could further illustrate the changes described.
- Figure 4 is informative, but the captions could be shortened and the gain/setting parameters noted for transparency. Also, clarify who performed the scans and whether inter-rater reliability was established.
- Consider adding a brief discussion on how the sonographic findings directly influenced adjustments in the rehabilitation plan.
Hip Joint Manipulation and Follow-Up
- Consider expanding on the theoretical basis for hip manipulation in this context and discuss any potential risks or contraindications.
- The rationale for hip manipulation is interesting but feels somewhat disconnected from the rest of the case. Connect this intervention more clearly to the patient’s residual instability or biomechanical deficits (including one or two key citations).
- Include objective measures (e.g., pre- and post-manipulation strength or stability tests) to support the patient’s subjective reports of improved stability.
Discussion
- Consider addressing the limitations of this single-case report, such as generalizability and the lack of long-term follow-up.
- The discussion is thorough but too long. Shortening the literature review, especially the sections summarizing prior case numbers, would enhance readability.
- Conclude with a clear summary of how to make real-time, ultrasound-guided rehab decisions.
- The hip manipulation subsection should tie back to the broader rehabilitation goals. How does this intervention fit into the long-term recovery plan?
Figures & Tables
- Table 1 effectively tracks progress, but exact p-values are unnecessary in a single-case report and can be omitted. Clarify whether ROM values are active or passive.
- Ensure all figures and tables are referenced in the text. The description of Figure 3 could be more detailed, such as including brace type and crutch use duration.
Conclusion
- The conclusion effectively summarizes the case but could highlight the broader clinical implications. For example, how might this approach be applied to other patients with similar injuries?
- Avoid repeating the abstract word-for-word. Instead, emphasize the most significant findings (e.g., the success of home-based rehabilitation).
References
- Citation density is good, yet a few statements lack immediate referencing (e.g., LLLT mechanism). Double-check the formatting style (journal abbreviations, DOI spacing) and remove duplicate entries.
- Check for consistency in reference formatting (e.g., some journal names are italicized, others not).
Author Response
Dear Editor and two Reviewers,
On behalf of our entire research team, we extend our gratitude for your invaluable assistance and guidance in refining our manuscript. Your insightful suggestions and expert feedback have been extremely helpful in enhancing the quality and clarity of our work. We have thoroughly addressed every query, incorporating substantial revisions that we believe significantly strengthen our paper. Once again, we express our appreciation for your dedicated review and time.
As a testament to our commitment to addressing your feedback thoroughly, we have enclosed two documents: the comprehensively revised manuscript, with modifications highlighted for easiness to find, and a comprehensive response document detailing our point-by-point responses to your insightful comments and recommendations.
Should there be any further points requiring our attention, we assure you that we remain fully committed to resolving any remaining issues to the best of our abilities. We are highly appreciative and grateful for the time and effort you have dedicated to our work.
Below you can find our point-by-point responses.
Thank you.
Kindest regards,
Emanuela Elena Mihai, M.D., Ph.D.
Reviewer 2:
Comments on the manuscript “Tailored Rehabilitation Program and Dynamic Ultrasonography after Surgical Repair of Bilateral Simultaneous Quadriceps Tendon Rupture in a Patient Affected by Gout: A Case Report” with manuscript ID: healthcare-3728528.
The manuscript presents a well-documented case of bilateral simultaneous quadriceps tendon rupture in a patient with gout, detailing the surgical repair, rehabilitation program, and dynamic sonography monitoring. The study is clinically relevant, especially given the rarity of such cases, and provides valuable insights into tailored rehabilitation approaches. However, some sections could benefit from clearer organization, more concise language, and additional details to enhance readability and scientific accuracy.
Abstract
- It would be helpful to include specific quantitative outcomes (e.g., ROM, VAS scores) to highlight the effectiveness of the interventions.
Author`s reply: Thank you so much for your input and recommendation. We addressed the evolution of these outcomes in the Abstract and you can find the following information highlighted in red in the comprehensively revised manuscript:
“The outcome measures included right and left knee range of motion (ROM), pain intensity measured on Visual Analogue Scale (VAS), functioning measured through ADL score, and gait assessment on Functional Ambulation Categories (FAC). All endpoints were measured at different time points, scoring significant improvement at discharge compared to baseline (e.g., ROM increased from 0 degrees to 95 degrees, while VAS decreased from 7 to 1, ADL score increased from 6 to 10, and FAC increased from 1 to 5).”
- Consider rephrasing "aiming to improve activities of daily living (ADL) and quality of life (QoL)" to "which aimed to improve ADL and QoL" for smoother flow.
Author`s reply: Thank you for pointing it out. To ensure a smoother flow, we made the changes accordingly in the manuscript.
“We showcase the clinical presentation, the surgical intervention, rehabilitation program, dynamic sonographic monitoring, and home-based rehabilitation techniques of this injury which aimed to improve activities of daily living (ADL) and quality of life (QoL).”
- Spell out abbreviations on first use (e.g., ADL, QoL).
Author`s reply: Thank you for your comment. We spelled out the abbreviations used for the first time both in the abstract and other sections of our manuscript accordingly.
“We report the case of a 68-year-old Caucasian male, afflicted with long-term gout who presented a simultaneous, bilateral quadriceps tendon rupture. We showcase the clinical presentation, the surgical intervention, rehabilitation program, dynamic sonographic monitoring, and home-based rehabilitation techniques of this injury which aimed to improve activities of daily living (ADL) and quality of life (QoL).”
- Add one sentence on objective ultrasound findings at follow-up to balance the functional outcomes already described.
Author`s reply: Thank you for your insightful comment. As recommended, we made the changes accordingly in the manuscript: “Musculoskeletal ultrasound findings showed mature tendon structure, consistent dynamic glide, and no scarring.”
Introduction
- The reason for presenting this case is well established. However, the introduction could be clearer in stating the manuscript’s objectives and explaining how it advances existing knowledge.
Author`s reply: Thank you for pointing this out. We made the changes in the manuscript as we aimed to be clearer in stating the manuscript`s objectives and what gaps we try to fill through our approach. Therefore, the following amendments were made:
“Given the scarcity of published reports addressing bilateral quadriceps tendon ruptures—especially those combining serial dynamic ultrasonography with a personalized, staged rehabilitation protocol—this study fills an important gap in the literature. Although several case reports and series describe surgical techniques or basic rehabilitation strategies, few offer a comprehensive approach that blends imaging follow-up with functional adaptation and early return to independence. This manuscript advances existing knowledge by demonstrating the utility of serial dynamic musculoskeletal ultrasound not only for diagnosis but also for monitoring healing progress and dynamically tailoring the rehabilitation process. Also, our approach is outlining a structured rehabilitation protocol informed by ultrasound findings, adaptable to the patient’s specific healing trajectory. Moreover, illustrating a home-based continuation strategy after discharge, is trying to reduce healthcare burden while maintaining functional recovery. Another tailored strategy is using joint manipulation which helps highlighting the role of this technique for muscle activation to ensure optimization of quadriceps muscle performance during recovery stage. Ultimately, this report contributes to improving clinical awareness, diagnostic accuracy, and functional outcomes in patients with rare, bilateral tendon injuries by promoting the integration of imaging and rehabilitation in a patient-specific, evidence-informed manner.”
- The epidemiology and rationale are well explained, but the opening paragraph contains many statistics in a single sentence, which makes it harder to read. Splitting it up and adding one or two recent references (post-2022) would improve the context.
Author`s reply: Thank you for your comment. We made the changes in the manuscript accordingly, we rewrote the paragraph from introduction and split it up to be easier to follow, and also added a new reference as per your suggestion:
“Quadriceps tendon rupture is a rare injury, typically affecting men over the age of 50, with a male-to-female ratio of 8:1 [1]. Its incidence is estimated at 1.37 per 100,000 persons per year, reinforcing the rarity of the condition [1-3].”
- The introduction offers a solid background on quadriceps tendon ruptures and their links to systemic conditions. However, the transition to the case report could flow more smoothly. Consider adding a sentence that explicitly highlights the novelty of this case, such as "This case is unique due to its bilateral nature and the use of dynamic sonography in rehabilitation."
Author`s reply: Thank you for your recommendation. We made the changes in the manuscript accordingly.
- The paragraph on ultrasonography is informative but could be shortened to reduce redundancy with later sections.
Author`s reply: Thank you for your input. According to your recommendation, we shortened the paragraph on ultrasonography.
“In addition to MRI, dynamic musculoskeletal ultrasound plays a key role in both diag-nosis and post-surgical monitoring, offering a non-invasive way to track tendon changes and muscle healing and guide individualized rehabilitation programs in real-time [10–12]. With high sensitivity (100% for quadriceps tendon, 87% for patellar tendon) and specificity, ultrasound is widely accessible in most clinical settings and complements clinical assessment effectively [10–12].”
Case Presentation
- Consider providing more information on the patient’s baseline functional status and comorbidities, as this would help contextualize the rehabilitation outcomes.
Author`s reply: Thank you for pointing this out. As per your recommendation, we provided more information on the patient`s baselines functional status and comorbidities:
“The personal medical and surgical history revealed type 2 diabetes treated with oral antidiabetic drugs, long-term gout, and hypertriglyceridemia. Patient`s body mass index (BMI) at the admission was 26,23 indicating overweight status. The patient denied taking any corticosteroid medication and did not have any prior history of trauma. His general health was stable under standard treatment for the aforementioned conditions.”
“The deep tendon reflexes could not be assessed due to the painful palpation of both knees. Patient baseline status was stable, but however, a high clinical suspicion of bilateral, simultaneous quadriceps tendon rupture was made during the clinical examination.”
- The clinical narrative is detailed and flows logically. You might shorten the transfer-of-care details to maintain momentum and include the patient’s body mass index, which is often important in tendon issues.
Author`s reply: Thank you for your comment. We shortened the details regarding the transfer and also included patient`s body mass index, as it is indeed an important factors in tendon lesions.
“Due to limited availability of operating theatres, the patient was transferred to the University Emergency Hospital in Bucharest, Romania. Upon admission, MRI (Figure 1) confirmed the clinical suspicion, revealing quadriceps tendon retraction.”
“Patient`s body mass index (BMI) at the admission was 26,23 denoting he was overweight.”
- The injury mechanism could be explained in more detail. For example, how did the misstep cause maximum knee flexion? A short biomechanical explanation would improve this section.
Author`s reply: Thank you for your insightful comment. As per your recommendation, we have made the following changes regarding the mechanism of injury and the biomechanical explanation.
“While descending stairs, the patient misstepped and collided with another person, leading to a loss of balance. In an attempt to regain his stability, he rapidly loaded his flexed knees eccentrically—a mechanism that placed excessive tensile stress on the quadriceps tendons while they were contracting to decelerate the fall—ultimately resulting in bilateral tendon rupture.”
- The MRI findings are clear, but the figure labels (A, B, C, D) are not referenced in the text. Refer to the figure when describing the images.
Author`s reply: Thank you for pointing this out. We referenced in the text the figure labels accordingly.
“Bilateral X-ray was performed the same day (Figure 1A and 1B).”
“Upon admission, MRI (Figure 1C and 1D) confirmed the clinical suspicion, revealing complete rupture of the quadriceps tendon at the level of the patellar insertion and quadriceps tendon retraction (Figure 1C and 1D). Additionally, the caption of the Figure 1, describes the finding for each image: Figure 1 A. Left knee lateral view on X-ray showing the low position of the patella. B. Right knee lateral view on X-ray showing the low position of the patella. C. Sagittal T2 left knee MRI image showing complete rupture of the quadriceps tendon at the level of the patellar insertion with 40 mm retraction. D. Sagittal T2 right knee MRI image showing complete rupture of the quadriceps tendon at the level of the patellar insertion with 24 mm retraction cranially.”
Surgical Treatment
- The surgical procedure is described in detail, but the rationale for choosing the Krackow suture over other techniques could be expanded. Referring to biomechanical studies (e.g., suture strength comparisons) would add depth to the discussion.
Author`s reply: Thank you for your insightful comment. We have already described in the Discussion section why the Krackow suture type was used :
“Even though there are various surgical techniques and sutures that can be used for the treatment of a tendon rupture, such as Bunnell, Kessler, or Krackow, the last one was used for our patient. The surgical team considered the biomechanical properties of the tendons and opted for the Krackow suture, whereas some studies have shown that the Krackow and the Bunnel type of sutures are superior to the Kessler suture when it comes to factors such as strength, endurance, and also durability [20,21]. Opting for a Krackow suture can be more challenging due to the technique, but this type of suture presented a more prominent axial resistance than the other types of sutures [22].” , but as per your request we have also added more details, while also referring to biomechanical studies to add more depth to the discussion:
“The Krackow technique was selected based on its superior biomechanical strength and resistance to cyclic loading, as demonstrated in comparative suture studies. This method provides enhanced tendon-to-bone fixation and minimizes gap formation during early rehabilitation phases, supporting a safer and more effective recovery.”
- Mention any intraoperative challenges or unexpected findings, if applicable, to provide a fuller picture of the surgical experience.
Author`s reply: Thank you for bringing this topic up. We mentioned a few of the challenges faced by the surgical team to provide a fuller picture of the surgical experience:
“Some challenges the surgical team had to consider were due to systemic conditions like diabetes and gout, as in the case of our patient, therefore, the possible degeneration of the tendons could have been an issue, making the tendon more friable and less able to hold sutures, but fortunately, the team successfully overcame these shortcomings. Additionally, given the bilateral involvement simultaneously, repairing both knees posed some ergonomic and logistical difficulties during the surgery, but the team managed to solve them effectively with a strict schedule and clear steps to be followed.
Another challenge was achieving equal tension and symmetry in this bilateral repair to prevent postoperative gait imbalance or extensor lag. Also, the correct positioning of anchors as SwiveLock in the patella must be very precise to ensure strong fixation while avoiding patellar fracture.”
- The phased rehabilitation program is comprehensive, but the timeline could be visualized in a table or flowchart for clarity.
Author`s reply: Thank you for your recommendation. We have created a table to be visualized more easily.
Table 1. Phased rehabilitation program
|
Phase |
Timeline |
Key Goals |
Interventions |
Measurements |
|
Phase 1 |
Weeks 1–4 |
Pain management, wound healing, surgical protection, maintain ROM, muscle strength, and overall function |
Knee braces locked in full extension; |
VAS, ROM, ADL, FAC |
|
Phase 2 |
Weeks 5–7 |
Regain motion, initiate muscle strengthening, cardiovascular rehab |
Braces (until week 6); |
VAS, ROM, ADL, FAC |
|
Phase 3 |
Weeks 8–9 |
Normalize gait, increase strength and ROM, regain ADL independence |
Progressive strengthening; |
VAS, ROM, ADL, FAC |
|
Phase 4 |
Week 10 onward |
Social and work reintegration, sport resumption |
Tailored home rehabilitation program; |
Follow-up ROM, ADL, return-to-work and non-contact sport activities |
“Abbreviations: ADL: activities of daily living; AROM: active range of motion; FAC: Functional Ambula-tion Categories; LLLT: low-level laser therapy; VAS: Visual Analogue Scale.*The structured rehabilitation program`s sessions were held 5 days per week, twice daily, during a period of 9 weeks. LLLT was applied once daily after surgical scars healing, for 6 weeks. The parameters used were the following: Wavelength: 808 nm; Power: 200 mW; Spot Size: 0.04 cm²; Energy per Point: 6 J.”
Post-Surgical Rehabilitation Program
- The phased program is a strength of the report. To enhance reproducibility, consider adding details on frequency and intensity for key exercises and clarifying the dosage of low-level laser therapy. A concise table with phase goals, precautions, and progression criteria would assist clinicians in replicating the regimen.
Author`s reply: Thank you for your insightful comment. We created a concise table (Table 1), incorporating details about rehabilitation phase and goals, frequency and intensity for key exercises, and also dosage for low-laser therapy.
“*The structured rehabilitation program`s sessions were held 5 days per week, twice daily, during a pe-riod of 9 weeks. LLLT was applied once daily after surgical scars healing, for 6 weeks. The parameters used were the following: Wavelength: 808 nm; Power: 200 mW; Spot Size: 0.04 cm²; Energy per Point: 6 J.”
Dynamic Sonography
- Including representative ultrasound images (if permitted) could further illustrate the changes described.
Author`s reply: Thank you for pointing this out. All the changes described in the ultrasound images (Figure 4) were done at different time points, according to the rehabilitation phase and are representative as they show how the tissues evolved from baseline to the end of the rehabilitation program. They were the most representative from a larger set. If there is allowed, we can add more pictures as supplemental material, to keep the main body of the manuscript as easier to follow as possible.
- Figure 4 is informative, but the captions could be shortened and the gain/setting parameters noted for transparency. Also, clarify who performed the scans and whether inter-rater reliability was established.
Author`s reply: Thank you for your comment. We clarified the aspects as presented below:
“Serial dynamic musculoskeletal ultrasound assessments were performed at admission, and at 6 and 9 weeks postoperatively, by a single experienced physiatrist to ensure consistency and eliminate inter-rater variability. Standard musculoskeletal ultrasound settings were applied, with the gain adjusted between 40–60 dB to optimize tendon and muscle visualization in both longitudinal and transverse views during dynamic assessment of the quadriceps tendons.”
- Consider adding a brief discussion on how the sonographic findings directly influenced adjustments in the rehabilitation plan.
Author`s reply: Thank you for your input. We have expanded this section accordingly on the significance of using ultrasonography and how these findings influenced the adjustments of the rehabilitation protocol. The following paragraph was added in the manuscript according to your recommendation:
“Musculoskeletal ultrasound examinations were conducted by a skilled physiatrist from our department, who was also responsible for the patient's admission. Alongside other members of the rehabilitation team, clinical findings and real-time ultrasound assessments were jointly reviewed to guide and adjust the rehabilitation protocol dynamically throughout the patient’s recovery. The reasoning behind dynamic musculoskeletal ultrasound is that this valuable tool is non-invasive, repeatable, and enables real-time feedback to adjust rehabilitation based on objective findings such as tendon continuity and healing status, structural integrity (e.g., fibrillar pattern, scarring), muscle activation during motion, tendon glide and contractility, effusion or post-surgical complications (e.g., hematoma, adhesions). All these features were taken into consideration for every step of the rehabilitation program and helped adapt the program in real-time, allowing a tailored intervention according to patient post-surgical and rehabilitation stage.”
Based on these findings alongside the clinical assessment, we started the phase 1 and the first part of phase 2 of the rehabilitation program. Moreover, after noting that the aspect was relatively homogeneous, with left and right echogenic quadriceps tendons with fibrillar pattern, several peritendinous calcifications and reduced subcutaneous tissue edema, we started the second part of phase 2, followed by phase 3 of the rehabilitation program as they are presented in Table 1.
Hip Joint Manipulation and Follow-Up
- Consider expanding on the theoretical basis for hip manipulation in this context and discuss any potential risks or contraindications.
Author`s reply: Thank you for your insightful comment. We have added more information accordingly:
“Given the patient’s residual deficits, particularly persistent quadriceps weakness and altered lower limb biomechanics, additional therapeutic strategies were explored. Six months post-surgery, a single high-velocity, low-amplitude (HVLA) hip manipulation was incorporated into the treatment plan to address compensatory pelvic alignment issues and facilitate neuromuscular activation of the quadriceps. Such targeted interventions are consistent with a global, individualized rehabilitation approach aimed at optimizing recovery, addressing biomechanical impairments, and promoting long-term stability and function.”
“This intervention was based on the concept of regional interdependence, whereby op-timizing hip joint mobility and lumbopelvic neuromotor control can indirectly enhance knee function and lower limb biomechanics. Hip manipulation has been shown to temporarily increase motor excitability and activation of muscles along the kinetic chain, particularly the vastus medialis and surrounding stabilizers of the knee [13]. In support of this, a recent systematic review and meta-analysis by Runge et al. found that adding manual therapy to exercise therapy significantly improves pain and functional outcomes in patients with knee and hip osteoarthritis—underscoring the neuromuscular and biomechanical benefits of these techniques beyond passive symptom relief [14]. However, clinicians must consider contraindications before applying joint manipulations. These include hip joint instability, osteoporosis, coagulopathies, previous hip arthroplasty, or any signs of metastatic involvement. In this case, thorough screening excluded these risks, and the manipulation was well tolerated without adverse effects.”
- The rationale for hip manipulation is interesting but feels somewhat disconnected from the rest of the case. Connect this intervention more clearly to the patient’s residual instability or biomechanical deficits (including one or two key citations).
Author`s reply: Thank you for your comment. In the text below we made the changes accordingly and added two new key citations:
“In this patient, the persistent biomechanical deficits—including poor single-leg stance control, delayed quadriceps recruitment, and compensatory trunk sway—warranted a proximal joint intervention to re-establish more optimal joint kinematics and neuromotor timing. By restoring hip mobility and reducing soft tissue restrictions, the manipulation likely enhanced the patient’s capacity for safe and symmetrical lower limb loading during gait and therapeutic tasks. Therefore, we proposed a physical therapy management using a single hip manipulation and therapeutic exercises six months after quadriceps tendon rupture surgery and re-habilitation program.
This intervention was based on the concept of regional interdependence, whereby op-timizing hip joint mobility and lumbopelvic neuromotor control can indirectly enhance knee function and lower limb biomechanics. Hip manipulation has been shown to temporarily increase motor excitability and activation of muscles along the kinetic chain, particularly the vastus medialis and surrounding stabilizers of the knee [13]. In support of this, a recent systematic review and meta-analysis by Runge et al. found that adding manual therapy to exercise therapy significantly improves pain and functional outcomes in patients with knee and hip osteoarthritis—underscoring the neuromuscular and biomechanical benefits of these techniques beyond passive symptom relief [14].”
- Include objective measures (e.g., pre- and post-manipulation strength or stability tests) to support the patient’s subjective reports of improved stability.
Author`s reply: Thank you for your comment. We added the following information in the comprehensively modified manuscript.
“Pre- and post-manipulation there were performed the following tests to assess patient`s strength and stability: Star Excursion Balance Test (SEBT) to evaluate dynamic postural control and proprioceptive stability given its usefulness in detecting unilateral or bilateral biomechanical deficits; Single-Leg Stance Time to assesses postural control, gluteal and quadriceps stability as improvement may correlate with perceived stability and neuromuscular reactivation; Timed Up and Go (TUG) to measure lower limb function and balance. Improvements after manipulation may reflect better neuromuscular efficiency.”
Discussion
- Consider addressing the limitations of this single-case report, such as generalizability and the lack of long-term follow-up.
Author`s reply: Thank you for your insightful feedback. As per your recommendation we added the following paragraph, also including strengths and limitations:
“This case report has several strengths and limitations as well. One strong feature is the comprehensive and individualized care: the case illustrates a well-structured, phase-based rehabilitation program integrating clinical, imaging, and functional assessments, including dynamic ultrasound monitoring, home-based rehabilitation, and joint manipulation. Additionally, the use of serial musculoskeletal ultrasound enabled real-time tracking of tendon healing, allowing tailored rehabilitation interventions based on healing progress. The integration of home-based rehabilitation ensured continuity of care and patient empowerment after discharge, therefore promoting functional independence and social reintegration. Another strength is the innovative inclusion of hip manipulation, as the case report contributes to emerging evidence suggesting that manual therapy may enhance neuromuscular activation, especially in the context of chronic muscle weakness. Limitations are related to the single-case design, which limits generalizability and outcomes may not apply to broader patient populations or different healthcare systems and medical contexts. Additionally, the follow-up only extended to six months after surgery, but long-term tendon integrity, return to sports, work, and quality of life may still suffer changes further this period of time. Also, other alternative treatments and interventions might have been suitable as well (e.g., continuous passive range of motion, neuromuscular reeducation devices) were not explored for comparison.”
- The discussion is thorough but too long. Shortening the literature review, especially the sections summarizing prior case numbers, would enhance readability.
Author`s reply: Thank you for your feedback. We have made several changes in the manuscript-discussion section to enhance readability and also streamlined the discussion.
“Some case reports showed both similar or slightly different results on ADLs [1,4,12].”
“Even though there are various surgical techniques and sutures that can be used for the treatment of a tendon rupture, the surgical team considered the biomechanical properties of the tendons and opted for the Krackow suture, whereas some studies have shown that the Krackow and the Bunnel type of sutures are superior to the Kessler suture when it comes to factors such as strength, endurance, and also durability [17,18].”
- Conclude with a clear summary of how to make real-time, ultrasound-guided rehab decisions.
Author`s reply: Thank you for your comment. We created a summary regarding real-time, ultrasound-guided rehabilitation decisions.
“In this case, dynamic musculoskeletal ultrasound was performed exclusively by a skilled physiatrist who also managed the patient’s care, ensuring consistency and continuity in assessment. Real-time ultrasound imaging allowed for objective monitoring of tendon healing, including continuity, fibrillar structure, muscle activation, tendon glide, and detection of complications such as hematomas or adhesions. These ultrasound findings, combined with clinical evaluation and informed ongoing rehabilitation decisions, enabled the interdisciplinary team to adapt the therapeutic plan in real-time. This individualized, responsive approach enhanced the precision and safety of each rehabilitation phase, optimizing patient recovery.”
- The hip manipulation subsection should tie back to the broader rehabilitation goals. How does this intervention fit into the long-term recovery plan?
Author`s reply: Thank you for pointing this out. We provided more information on how this intervention fit into the long-term recovery plan, and tied it within the broader rehabilitation goals.
“The long-term rehabilitation goals for this patient following bilateral quadriceps tendon rupture surgery were centered on restoring lower limb strength, functional independence, gait stability, and reintegration into daily and occupational activities. These objectives were pursued through a structured inpatient and home-based rehabilitation program, closely guided by serial dynamic musculoskeletal ultrasound assessments to tailor interventions in real-time. Given the patient’s residual deficits, particularly persistent quadriceps weakness and altered lower limb biomechanics, additional therapeutic strategies were explored. Six months post-surgery, a single high-velocity, low-amplitude (HVLA) hip manipulation was incorporated into the treatment plan to address compensatory pelvic alignment issues and facilitate neuromuscular activation of the quadriceps. Such targeted interventions are consistent with a global, individualized rehabilitation approach aimed at optimizing recovery, addressing biomechanical impairments, and promoting long-term stability and function.”
Figures & Tables
- Table 1 effectively tracks progress, but exact p-values are unnecessary in a single-case report and can be omitted. Clarify whether ROM values are active or passive.
Author`s reply: Thank you for you comment. As you pointed out, Table 1 (now table 2) has the purpose to show the track of progress and has no p-values, as they are unnecessary in a single case report. We additionally clarified that the ROM values were active.
- Ensure all figures and tables are referenced in the text. The description of Figure 3 could be more detailed, such as including brace type and crutch use duration.
Author`s reply: Thank you for pointing this out. We added the information in the manuscript. Regarding the crutches, it was mentioned that the patient used them for 8 weeks.
“Figure 3. A. Patient wearing a bilateral functional brace (unhinged ROM knee brace adjustable with extended support) for 6 weeks to keep the lower limbs in full extension. B. Patient walking with the aid of crutches at 8 weeks on even surface and without crutches at discharge (9 weeks).”
Conclusion
- The conclusion effectively summarizes the case but could highlight the broader clinical implications. For example, how might this approach be applied to other patients with similar injuries?
Author`s reply: Thank you for your comment. We aimed to highlight the broader clinical implications and how this strategy might be applied to patients facing similar injuries, as per your recommendation:
“This tailored and comprehensive approach emphasizes the importance of addressing biomechanical compensations that may persist despite surgical repair and rehabilitation programs. The comprehensive management provides a valuable framework that can be extended to similar complex musculoskeletal injuries and broader clinical integration of such individualized protocols may improve long-term outcomes, restore functional independence, and enhance quality of life in active patients.”
- Avoid repeating the abstract word-for-word. Instead, emphasize the most significant findings (e.g., the success of home-based rehabilitation).
Author`s reply: Thank you for your insightful feedback. We rewrote the conclusion and aimed to emphasize the most significant changes as per your recommendation:
“This case underscores the value of a timely surgical intervention combined with a structured, multidisciplinary rehabilitation strategy for the recovery from bilateral quadriceps tendon rupture. The successful use of dynamic musculoskeletal ultrasound throughout the rehabilitation process enabled personalized, real-time adjustments that supported tendon healing and functional progression. Home-based rehabilitation played a crucial role in maintaining gains post-discharge and facilitating safe reintegration into social and professional life. Importantly, the inclusion of a targeted single hip manipulation as part of the broader rehabilitation program contributed to enhanced quadriceps muscle activation and neuromechanical stability, complementing the patient’s recovery trajectory. This tailored and comprehensive approach emphasizes the importance of addressing biomechanical compensations that may persist despite surgical repair and rehabilitation programs. The comprehensive management provides a valuable framework that can be extended to similar complex musculoskeletal injuries and broader clinical integration of such individualized protocols may improve long-term outcomes, restore functional independence, and enhance quality of life in active patients.”
References
- Citation density is good, yet a few statements lack immediate referencing (e.g., LLLT mechanism). Double-check the formatting style (journal abbreviations, DOI spacing) and remove duplicate entries.
Author`s reply: Thank you for pointing this out. We changed accordingly in the manuscript, checked the formatting style and any duplicate entries.
- Check for consistency in reference formatting (e.g., some journal names are italicized, others not).
Author`s reply: Thank you for pointing this out. We carefully and thoroughly checked the reference formatting for consistency.

Round 2
Reviewer 2 Report
Comments and Suggestions for Authors
I sincerely thank the authors for their thoughtful responses and the thorough revisions made to the manuscript. These changes have significantly enhanced the clarity and overall quality of the work.
Author Response
Dear Reviewer,
On behalf of our entire research team, we extend our gratitude for your invaluable assistance and guidance in refining our manuscript entitled “Tailored Rehabilitation Program and Dynamic Ultrasonography after Surgical Repair of Bilateral Simultaneous Quadriceps Tendon Rupture in a Patient Affected by Gout: A Case Report”.
Kindest regards,
Emanuela Elena Mihai, M.D., Ph.D.
Reviewer 2 Comments: I sincerely thank the authors for their thoughtful responses and the thorough revisions made to the manuscript. These changes have significantly enhanced the clarity and overall quality of the work.
Author`s reply: We would like to express our gratitude towards the reviewer for helping us improve our manuscript and enhance its overall quality.